# Association of Dietary Patterns with Cardiovascular and Kidney Phenotypes in an Uruguayan Population Cohort

**DOI:** 10.3390/nu13072213

**Published:** 2021-06-27

**Authors:** Paula Moliterno, Carmen Marino Donangelo, Luciana Borgarello, Matías Pécora, Alicia Olascoaga, Oscar Noboa, José Boggia

**Affiliations:** 1Departmento de Nutrición Clínica, Escuela de Nutrición, Universidad de la República, Montevideo CP 11600, Uruguay; pmoliterno@nutricion.edu.uy; 2Escuela de Nutrición, Universidad de la República, Montevideo CP 11600, Uruguay; cmarino@nutricion.edu.uy; 3Laboratorio de Patología Clínica, Universidad de la República, Montevideo CP 11600, Uruguay; luborga@hotmail.com (L.B.); aolascoa@gmail.com (A.O.); 4Departmento de Fisiopatología, Universidad de la República, Montevideo CP 11600, Uruguay; pecora.matias@gmail.com; 5Centro de Nefrología, Universidad de la República, Montevideo CP 11600, Uruguay; onoboa@gmail.com

**Keywords:** dietary patterns, cardiovascular phenotypes, kidney phenotypes, population science, principal component analysis

## Abstract

The impact of habitual diet on chronic diseases has not been extensively characterized in South America. We aimed to identify major dietary patterns (DP) in an adult cohort in Uruguay (Genotype Phenotype and Environment of Hypertension Study—GEFA-HT-UY) and to assess associations with metabolic, anthropometric characteristics, and cardiovascular and kidney phenotypes. In a cross-sectional study (*n* = 294), DP were derived by the principal component analysis. Blood and urine parameters, anthropometrics, blood pressure, pulse wave velocity, and glomerular filtration rate were measured. Multivariable adjusted linear models and adjusted binary logistic regression were used. Three DP were identified (Meat, Prudent, Cereal and Mate) explaining 22.6% of total variance in food intake. The traditional Meat DP, characterized by red and barbecued meat, processed meat, bread, and soft drinks, was associated with worse blood lipid profile. Prudent DP, characterized by vegetables, fish, and nuts, and lower loads for bread and crackers, was associated with reduced risk of vitamin D deficiency. Cereal and Mate DP, was characterized by higher loads of cereals, bread, and crackers, and mate infusion, with higher odds of excessive body weight. No direct associations of dietary patterns with hypertension, arterial stiffness, chronic kidney disease, and nephrolithiasis were found in the studied population, nor by age categories or sex.

## 1. Introduction

Noncommunicable diseases (NCDs) are the most prevalent cause of death globally [1]. In most countries of Latin America, cardiovascular risk factors are highly prevalent [2], and particularly those related to unhealthy lifestyles and dietary patterns [3]. Although aging plays an important and unavoidable role in the development of NCDs, sustained exposure to unhealthy diets fastens physiological and morphological changes leading to early vascular aging and premature advancement of disease [4]. Notably, a recent world-wide report has highlighted the impact of suboptimal diet in mortality and disability-adjusted life-years (DALYs) [5]. The report estimated 11 million deaths and 255 million DALYs attributable to dietary risk factors in 2017 [5]. Uruguay is a small (176,000 square kilometers) country in the southeastern region of South America, homing an estimated 3.5 million people. According to the Uruguayan Ministry of Health surveys [6,7], the prevalence of NCDs in the adult population rose between 2006 and 2013 probably due to population aging and changes in lifestyle patterns [6,7]. Between 2006 and 2013, the prevalence of overweight and obesity increased from 57 to 65% from 2006 to 2013, hypertension from 30 to 39%, and diabetes from 5 to 8%. Moreover, 30% of the adult population is physically inactive [6,7]. Regarding dietary habits of Uruguayan adults, the estimated daily energy intake per capita (2432 kcal) exceeds requirements by 19% [8], around 90% of adults consume inadequate amounts of vegetables and fruits [7], salt intake is high [9], and consumption of highly processed foods have had an outstanding rise [10]. Therefore, dietary habits in Uruguay may pose a health concern.

Dietary patterns (DP) have been widely explored to study the diversity in diet, as they represent a more comprehensive description of dietary intake [11]. This approach that studies combinations of foods and beverages in the diet rather than specific nutrients, enables capturing synergistic or cumulative effects of foods on the relationship of diet and metabolic and functional markers of disease [12]. To the best of our knowledge, there are no studies characterizing different DP in Uruguay and assessing the association between a typical Uruguayan DP and cardiovascular and kidney risk phenotypes. Defining local DP will contribute to stepping up national nutrition policies by down-modulating intermediate risk factors and prevention of cardiovascular risk phenotypes and disease.

The aim of the present study was to identify major DP in an adult cohort in Uruguay (Genotype Phenotype and Environment of Hypertension Study—GEFA-HT-UY) and to assess associations with metabolic, anthropometric characteristics, and cardiovascular and kidney phenotypes.

## 2. Materials and Methods

### 2.1. Population

The GEFA-HT-UY study is a prospective cohort with a sample of 572 participants, that started in April 2012 randomly recruiting nuclear families from the inhabitants of a geographically defined area located approximately 10 km from downtown Montevideo [13]. The sample does not intend to be nationwide representative. The cohort included family members older than 18 years and without an upper age limit. Participants were invited by telephone and home visits. The participation rate among eligible subjects at recruitment was 72.7%. Examinations were undertaken at a health center located within the neighborhood where trained interviewers administered questionnaires inquiring into each participant’s medical history, smoking, drinking, and dietary habits, as well as use of medication. The ethics committee of the University Hospital (Hospital de Clínicas, Dr Manuel Quintela, Universidad de la República, Uruguay) approved the study protocol (3 January 2011), and all the participants gave an informed written consent. Our study included all the participants from the GEFA-HT-UY study that provided baseline dietary information. For the present analysis, we excluded 277 participants who did not provide information on diet or reported extreme daily energy intake (<800 or >4000 kcal/day), and one participant who had an incomplete cardiovascular assessment. Finally, 294 participants (51.4%) were included in the analysis (Figure 1). Subjects included and not-included in the present analysis did not differ in demographic characteristics (*p* > 0.09), except for that they had higher body mass index (29.5 ± 6.4 and 28.2 ± 5.2 kg/m^2^) and pulse wave velocity (8.9 ± 2.9 and 7.6 ± 1.9 m/s); higher percentage of obesity (39.5 and 23.1%), arterial stiffness (25.3 and 4.6%), and women (67.7 and 58.0%); and lower percentage of chronic kidney disease (7.1 and 19.2%, respectively) (*p* < 0.05).

#### 2.1.1. Dietary Assessment and Food Grouping

Dietary information was collected through an interview by a trained professional using an adapted food frequency questionnaire (FFQ) [14]. The usual intake (daily, weekly or monthly) of 130 single food items and serving size, during the 12 months before the examination were assessed. A photographical food atlas was available to best estimate the portion size of each food item [15].

For DP analysis, food items were aggregated into 27 separate food groups considering nutrient profile or culinary typical practices (e.g., meat cooked by barbecue was separated from other meats since it is commonly consumed in Uruguay). The groups were: Barbecued meat, red meat, processed meat, fish, white meat (poultry and pork), cereal (including rice, noodles, corn flour), bread and crackers, snacks (chips), dipping sauces and instant soups, leafy vegetables, other vegetables, eggs, nuts, fruits, dairy (milk and yoghurt), cheese, wholegrain cereals, potatoes, nutritional supplements, vegetable oil, sugar and marmalade, cakes and pastries, butter and margarine, soft drinks, alcohol drinks, coffee and energy drinks, and “mate” (a traditional infusion prepared from leaves of *Ilex paraguariensis*).

Daily food and beverage intakes (g or mL/day) were calculated from the frequency of intake and portion size of each food item or beverage consumed. The total nutrient intake was estimated as the sum of each food contribution to the nutrient according to the portion size ((∑ portion size/day) × nutrient content). Energy and nutrient intakes were estimated using the System Analysis and Food Registry database, [16] and the US Department of Agriculture’s National Nutrient Database for Standard Reference [17]. In addition, for dairy products the national specific composition information was used in order to obtain a more accurate composition estimation. Energy, protein, carbohydrates, lipids (including saturated fatty acids (SFA), monounsaturated (MUFA), and polyunsaturated fatty acids (PUFA)), cholesterol, dietary fiber, calcium, and sodium intakes from foods and beverages were estimated.

#### 2.1.2. Dietary Pattern Analysis

The principal component analysis (PCA) was conducted to derive DP using the SAS PROC FACTOR. To increase the proportion of variance explained by the factors, food information was categorized in 27 food items (in g or mL) to perform the analysis. The sample size of 294 allowed for examination of 27 food groups using PCA, as 10 participants per variable are required for robust results [18]. Sampling adequacy was tested by estimating the Kaiser-Meyer-Olkin index (0.62) and Bartlett test of sphericity (<0.0001), resulting in an adequate sample size [19]. After a first exploratory analysis, we defined the major factors (DP) to be retained based on the following criteria: The proportion of variance explained by each factor considering only eigenvalues >1, inflection point of the curve, and interpretability after varimax orthogonal rotation, which leads to uncorrelated factors that are considered easier to infer [20]. For each factor (DP) retained, each food item received a factor loading, which represents the correlation coefficient between the food item and the factor (DP). Afterwards, each factor (DP) was named following preferably a quantitative criteria [21], by considering the first three food items with significant and positive loads to the factors. Post-rotation loads of ±0.3 (or of higher magnitude) were considered significant [20,21]. The DP analysis was made considering females and males in a mixed sample, as previous studies have found similar DP across sex groups when DP were derived separately for women and men [21]. Relationships with sex were afterwards explored through other analyses.

### 2.2. Physical Activity

The frequency and average duration of each type of physical activity were derived from standardized data collected on questionnaires. To estimate the total energy expenditure (TEE) from physical activity, separate metabolic equivalent (MET) minutes per week were calculated for each activity according to the following formulas: MET coefficient of activity, * duration (minutes per time), and * frequency (times per week) [22]. Physical inactivity was considered for TEE: <600 MET- min/week.

### 2.3. Anthropometric Measurements

Trained technicians measured body height to the nearest 0.5 cm using a pliable measurer (Seca, Hamburg, Germany) with the participant standing against a wall and maintaining the head in the Frankfort Horizontal Plane position. Body weight measurements (HBF 415, Omron, Japan) were made to the nearest 100 g with the participant wearing light indoor clothing without shoes.

The body mass index (BMI) was calculated as weight (kg) divided by square height (m^2^) and classified as underweight (<18.5 kg/m^2^), normal (18.5–24.9 kg/m^2^), overweight (25.0–29.9 kg/m^2^) or obese (≥30 kg/m^2^). The waist circumference measurement was made using an inelastic measuring tape (Seca, Hamburg, Germany) in the midpoint between the top of the iliac crest and the lower margin of the last palpable rib in the mid axillary line. We calculated the waist-to-height ratio as waist circumference (cm) divided by height (cm) as a predictive indicator of early health risks associated with central obesity [23].

### 2.4. Blood Samples and Biochemical Measurements

Venous blood samples were obtained in the morning after 12 h of fasting and were kept at 4 °C. Within a 2 h period, samples were processed for biochemical analysis of serum total cholesterol, triglycerides (TG), low and high-density lipoproteins cholesterol (LDLc, HDLc), fasting glucose and insulin, serum 25-hydroxy-vitamin-D (25(OH)D), and blood neutrophils and lymphocytes. Routine blood and serum analyses were performed using Cobas-6000 (Roche, Mannheim Germany). Serum 25(OH)D was determined by chemiluminescence (Elecsys vitamin D total II, Cobas; Roche Diagnostics, Mannheim, Germany).

The TG/HDLc ratio was calculated as an index of cardiovascular risk [24]. The homeostasis model assessment of insulin resistance (HOMA-r) was calculated as (fasting insulin (μU/mL) × fasting glucose (mg/dL)/405) to classify the cardiometabolic risk, as insulin resistance has been associated to metabolic alterations and inflammation [25]. The neutrophils/lymphocytes ratio was calculated by dividing neutrophils by lymphocytes, as a marker of subclinical inflammation [26].

Diabetes mellitus was diagnosed as the fasting glucose level of ≥126 mg/dL (7 mmol/L) or use of antidiabetic drugs.

### 2.5. Blood Pressure Measurements

Blood pressure (BP) was measured by trained technicians, using an auscultatory technique with mercury sphygmomanometers and following standardized protocols. Yearly quality controls were performed using the British Hypertension Society video. After the participants had rested for 5 min in a sitting position, trained observers obtained five consecutive BP readings (phase I systolic pressure and phase V diastolic pressure) to the nearest 2 mmHg. Standard cuffs had a 12 × 24 cm inflatable portion, however, if the upper arm girth exceeded 31 cm, larger cuffs (15 × 35 cm bladders) were used. The five BP readings were averaged for the analysis.

Hypertension was defined according to the European and regional guidelines as a brachial BP of at least 140 mmHg systolic or 90 mmHg diastolic or the use of antihypertensive drugs. The peripheral pulse pressure was the difference of systolic (SBP) minus diastolic blood pressure (DBP) derived from the brachial BP measurement. The uncontrolled hypertension was analyzed through office BP and defined as a brachial BP of at least 140 mmHg SBP or 90 mmHg DBP, and by the ambulatory blood pressure monitoring (ABPM) measurement as a BP of at least 130 mmHg SBP or 80 mmHg DBP.

### 2.6. Arterial Ageing Parameters

Increased arterial stiffness (AS) is already recognized as an index of early vascular ageing. We computed the pulse pressure as already described above, and obtained aortic pulse wave velocity as a surrogate marker of arterial stiffness. After the participants had rested for 15 min in the supine position, we sequentially recorded the right carotid and right femoral waveforms by applanation tonometry. We used a high-fidelity SPC-301 micromanometer (Millar Instruments, Inc., Houston, TX, USA) interfaced with a computer running SphygmoCor software, version 8.2 (AtCor Medical Pty. Ltd., West Ryde, New South Wales, Australia). Aortic pulse wave velocity was measured by sequential ECG-gated recordings of the arterial pressure waveform at the carotid and femoral arteries. Distances from the suprasternal notch to the carotid sampling site (distance A), and from the suprasternal notch to the femoral sampling site (distance B) were measured. The pulse wave travel distance was calculated as distance B minus distance A. The pulse transit time was the average of 10 consecutive beats. Pulse wave velocity was the distance in meters divided by the transit time in seconds. A cut-off value of 10 m/s was used to discriminate normal arterial elasticity and arterial stiffness.

### 2.7. Assessment of Kidney Parameters

Serum creatinine was measured by modified kinetic Jaffé methods (COBAS, Roche diagnostics, Mannheim, Germany). The detection limit is 0.17 mg/dL, and the coefficient of variation was 1.6%. We used the creatinine method that has calibration traceable to an IDMS reference measurement procedure according to the present recommendations [27]. The urinary albumin to the creatinine ratio (ACR) was calculated. We estimated the glomerular filtration rate (eGFR) using the chronic kidney disease epidemiology collaboration (CKD-EPI) equation [28]. The chronic kidney disease (CKD) was defined as an eGFR < 60 mL/min/1.73 m^2^ or an ACR > 30 mg/g based on a single determination. Nephrolithiasis history was assessed using a standardized questionnaire. For analysis, we computed eGFR as a continuous variable and CKD and nephrolithiasis as categorical variables.

### 2.8. Statistical Methods

The SAS software, version 9.4 (SAS Institute, Cary, NC, USA) was used for database management and statistical analysis.

Continuous variables were expressed as the mean ± standard deviation (SD), and categorical variables were expressed as the absolute number and proportions. Means and proportions were compared using a Student’s *t*-test and a Chi squared test, respectively.

The identification and labelling of DP were done as described in a previous section. For each participant, a factor score for each DP was calculated, as a linear composite of the optimally weighted food items by factor loadings. The factor score was used to categorize individuals according to the “level of adherence” to each DP (low, medium, and high) as tertiles of DP load. The lowest tertile represents the participants with poor adherence to the DP and the highest tertile represents the participants with the best adherence. Trend tests were performed to assess the pattern of the relationship between tertiles of DP and general characteristics of participants and dietary components, using PROC REG (linear regression) for continuous variables and PROC FREQ statement with the TREND option for binary categorical variables.

To assess the associations between DP and metabolic, anthropometric, and cardiovascular and kidney phenotypes, multivariable linear regression models were used to estimate by tertile of DP load, the adjusted changes of each outcome variable relative to the lowest tertile. Models were adjusted for BMI, age, and energy daily intake as continuous variables, and for sex, smoking status (current smoking: Yes/no), physical activity (inactive/active), and education (≤9/>9 years) as categorical variables. For serum 25(OH)D and BP the models were additionally adjusted for season (summer, autumn, winter, spring), and use of antihypertensive drugs (yes/no), respectively. The analyses were also performed by categories of age, considering younger and older participants according to the median age of the whole population.

The adjusted odds ratio (OR) was derived using the binary logistic regression PROC LOGISTIC to assess the association of identified DP and cardiovascular and kidney risk phenotypes. Before performing logistic regression, participants were classified according to cardiovascular risk phenotypes as follows: Hypertension and uncontrolled hypertension (see Blood Pressure Measurement Section); high TG/HDLc ratio (>3.75 for men and >3.0 for women, according to recommendations for normal fasting TG (<150 mg/dL) and HDLc (≥40 mg/dL for men and ≥50 mg/dL for women); overweight (see Anthropometric Measurements Section); overweight and obesity together (see Anthropometric Measurements Section); arterial stiffness (see Arterial Ageing Parameters Section); high waist-to-height ratio (≥0.5) [23]; vitamin D deficiency (serum 25(OH)D <12 ng/mL) [29]; and CKD (see Assessment of Kidney Phenotypes Section). The models tested were adjusted as previously mentioned. ORs were assessed in the whole group, by age categories, and by sex.

After stratification for sex, we interpolated the missing values of body mass index (*n* = 9), blood lipids (*n* = 14), office blood pressure (*n* = 9), and serum 25(OH)D (*n* = 23) from the regression slope on age [30]. Statistical significance was considered when *p* < 0.05 on two-sided tests.

## 3. Results

### 3.1. Characteristics of Participants

The 294 participants included 199 (67.7%) women. The mean age was 53 years for women and 51 years for men. Of the 294 participants, 107 (36.4%) were hypertensive, 31 (10.5%) had diabetes, 37 (12.6%) reported nephrolithiasis, 55 (20%) were current smokers, 222 (75.5%) were physically inactive, and 97 (35.3%) reported drinking alcohol at least once a week. Of the hypertensive patients, 82 (76.6%) were using antihypertensive drugs. Overweight and obesity were present in 221 (75.2%) of participants and mean BMI was 29.5 ± 6.4 kg/m^2^, with no difference between sex (*p* = 0.57). The average participant waist-to-height ratio was 0.59 ± 0.092, fasting glucose was 97.3 ± 26.3 mg/dL, total cholesterol was 205.7 ± 42.4 mg/dL, neutrophils-to-lymphocytes ratio was 1.77 ± 0.81, 25(OH)D was 24.4 ± 9.7 ng/mL, and eGFR was 95.4 ± 20.6 mL/min/1.73m^2^. The average office SBP/DBP was 124.6 ± 17.6/79.6 ± 9.9 mmHg. Characteristics of participants by age and sex are shown in the Appendix A).

### 3.2. Dietary Patterns and Characteristics of Participants

Three distinct DP were derived through PCA accounting for 22.6% of the variance in food intake. They were named as “Meat”, “Prudent”, and “Cereal and Mate” DP taking into consideration their major food groups and nutritional characteristics. The factor loadings for each of the identified DP and the matrix plot are shown in Table 1 and Figure 2, respectively. Higher loads for red and processed meat, soft drinks, as well as barbecued meat characterized the Meat DP. The Prudent DP had higher contributions from vegetables, fish, and nuts, representing a healthier food-selection, with lower loads for bread and crackers. The Cereal and Mate DP was characterized by higher loads for cereals, bread, and crackers, and for mate, an herbal infusion commonly consumed among Uruguayans, while had lower contributions of dairy foods (Table 1).

Dietary intakes of participants according to tertiles of DP scores are shown in Table 2. Participants in the highest tertile of the Meat DP had higher intakes of total energy, protein, sodium, and alcohol (*p* < 0.0001), and lower intakes of fiber, MUFA, and PUFA (*p* < 0.04) compared to those in tertile 1 (reference). Participants in the highest tertile of the Prudent DP had higher intakes of protein, fat (particularly MUFA), fiber, and calcium (*p* < 0.01), and lower intakes of carbohydrate and alcohol (*p* < 0.005). For the Cereal and Mate DP, participants in the highest tertile had higher intakes of total energy, carbohydrate, and sodium (*p* < 0.002), and lower intakes of protein, fat, PUFA, and calcium (*p* < 0.02), compared to those in the lowest tertile.

Demographic and anthropometric characteristics, conventional risk factors, and cardiovascular and kidney phenotypes by tertiles of DP are summarized in Table 3. Compared to the lowest tertile, participants in the highest tertile of the Meat DP and Cereal and Mate DP had a higher (*p* < 0.05) proportion of current smokers (33.3 vs. 7.5% and 27.5 vs. 12%, respectively) and alcohol use (47.8 vs. 30.1% and 47.3% vs. 31.2%, respectively) compared to the lowest tertile. Participants in the highest tertile of the Meat DP were less likely to be women (*p* = 0.068) and were younger (*p* < 0.0001). On the other hand, participants in the highest tertile of the Prudent DP were older (*p* = 0.018) compared to those in the lowest tertile.

Participants with lower adherence to the Prudent DP were less educated (70.1 vs. 55.1%) compared with the highest tertile (*p* = 0.029), whereas participants with higher adherence to the Cereal and Mate DP were less educated (70.7 vs. 53.6%) and more physically inactive (79.8 vs. 65.3%) than those in the lowest tertile (*p* < 0.05). Only in the Meat DP, the proportion of hypertension, uncontrolled hypertension by office measurement, arterial stiffness, and diabetes was lower across tertiles of DP (*p* < 0.05) (Table 3). The participants’ characteristics according to tertiles of DP scores by age categories are shown in the Appendix A.

### 3.3. Metabolic, Anthropometric, Cardiovascular, and Kidney Variables by Tertiles of Dietary Patterns

Participants in the highest tertile of the Meat DP had significantly higher triglycerides (median adjusted difference 16.68 mg/dL; *p* < 0.04) and lower HDLc (median adjusted difference −3.48 mg/dL; *p* < 0.012), compared to participants in the lowest tertile (Table 4).

Thus, the TG/HDLc ratio was significantly higher (median adjusted difference 0.70; *p* < 0.0031) in participants with highest adherence to the Meat DP compared to those in the lowest tertile. Similar results were observed for HOMA-r (median adjusted difference 0.54; *p* < 0.044). However, increasing adherence to the Prudent DP and the Cereal and Mate DP was not associated with significant changes in the selected cardiovascular and kidney risk factors (Table 4).

A sub-analysis performed according to the age categories (Appendix A) showed that younger participants (≤54 years) in the highest tertile of the Meat DP had significantly higher TG (median adjusted difference 22.19 mg/dL; *p* < 0.036), lower HDLc (median adjusted difference −3.61 mg/dL; *p* < 0.047), and higher proteinuria (median adjusted difference 0.038 g/24 h; *p* < 0.0070) compared to participants in the lowest tertile. In contrast, older participants (>54 years) in the highest tertile of the Meat DP had significantly lower 25(OH)D levels (median adjusted difference −2.94 ng/mL; *p* < 0.012) compared to participants in the lowest tertile. Only younger participants with highest adherence (top tertile) to the Prudent DP had higher serum creatinine and lower eGFR compared to those with lower adherence (lowest tertile). Moreover, younger participants in the highest tertile of the Cereal and Mate DP had higher HDLc (median adjusted difference 5.01 mg/dL; *p* < 0.024), office SBP (median adjusted difference 5.09 mm Hg; *p* < 0.022), systolic ABPM (median adjusted difference 10.74 mm Hg; *p* < 0.0004), and diastolic ABPM (median adjusted difference 6.82 mm Hg; *p* < 0.0046) measurement, and albumin to creatinine ratio (median adjusted difference 11.44 mg/g; *p* < 0.0005) compared to younger participants in the lowest tertile (Appendix A). In contrast, in older participants, there was no association between increasing adherence to either DP and selected cardiovascular and kidney risk phenotypes, except for vitamin D (Appendix A).

Adjusted OR and 95% confidence intervals (CI) for some selected cardiovascular risk phenotypes are summarized across tertiles of DP (Table 5). Highest adherence to the Prudent DP was associated with reduced odds of vitamin D deficiency (serum 25(OH)D < 12 ng/mL) (OR = 0.22; 95% CI: 0.048–0.99, *p* = 0.049), while highest adherence to the Cereal and Mate DP was associated with higher odds of excessive weight (OR = 2.48; 95% CI: 1.13–5.43, *p* = 0.023). A sensitivity analysis assessing those variables as continuous showed no associations (Appendix A). When ORs were performed according to the age categories (Appendix A), only in older participants, highest adherence to the Cereal and Mate DP was associated with higher odds of overweight (OR = 2.84; 95% CI: 1.11–7.27, *p* = 0.030), while highest adherence to the Prudent DP was associated with lower odds of vitamin D deficiency (OR = 0.063; 95% CI: 0.005–0.75, *p* = 0.029). When ORs were performed for both sexes, separately (Appendix A), men with highest adherence to a Cereal and Mate DP had higher odds for overweight (OR = 6.57; 95% CI: 1.99–21.67, *p* = 0.0020) but no association was found when OR was performed for overweight and obesity together.

## 4. Discussion

In this cross-sectional study conducted in Uruguayan adults, three DP were identified (Meat, Prudent, Cereal and Mate). Our study indicated that the highest adherence to the Meat DP was associated with higher TG, HDLc, TG/HDLc ratio and HOMA-r, particularly in younger participants. The Prudent DP was associated with reduced risk of vitamin D deficiency, particularly in older participants. Following the Cereal and Mate DP was associated with increased risk of excessive body weight. To the best of our knowledge, this is the first study to characterize different DP in adults in Uruguay and to assess the association between a typical Uruguayan DP and cardiovascular and kidney phenotypes.

The DP identified in the present study were consistent with other previously described DP in the region [31,32,33,34]. The Meat DP identified in our study consisted of foods that characterize Uruguayan eating habits such as red and barbecued meat, bread, and soft drinks. This pattern resembles other previously DP described in Uruguay and Argentina with denominations such as “Traditional” or “Western” [33,35,36]. The Cereal and Mate DP emerged as another traditional local DP not previously described, with mate infusion beverage as an important component, together with starchy foods such as cereals, bread, and crackers. Despite being widely consumed in Uruguay, the previously identified DP were not characterized by mate beverage intake [33,35]. In studies conducted in Argentina and Brazil [32,36], the intake of mate beverage emerged with a positive load as part of a “Traditional” [36] DP or with a negative load for the “Healthy” [32] DP. Finally, the Prudent DP in our study emerged as the DP with a healthier selection of foods, with vegetables, fish, and nuts having higher contributions, while bread and crackers had lower loads. DP with similar nutritional components were also previously denominated in the region as “Prudent” [31], as well as “Healthy” [32,34,35].

### 4.1. Dietary Patterns and Metabolic Characteristics

In our study, the Meat DP was associated with a blood lipid profile that may result in insulin resistance and dyslipidemia [24]. Accordingly, a higher HOMA-r, as another predictor of insulin resistance was found in those with highest adherence to the Meat DP. In contrast to our study, previous reports in the region did not find an association between the highest score in the “Caloric” DP (meat, processed meat, sweet and sugar, and soft beverages, and artificial juices) [32] or the “Western” DP (red meat, eggs, snacks, pastry, cakes, and refined grains) [31], and lipid biomarkers. Fiber and unsaturated fatty acids intake were lower in those at the top tertile compared to the lowest Meat DP tertile, which may be related to the serum lipoproteins phenotype, as diets with a low fiber content have shown to increase more TG compared to fiber-rich foods [37]. Other meat-characterized DP described in the region also showed lower fiber intake with highest adherence, related to the quality of the food selection [31,32]. In our study, a higher alcohol intake in those with higher adherence to the Meat DP may also be related to the higher TG levels, as alcohol increases the production and levels of very low-density lipoproteins cholesterol (VLDLc) [37], particularly in overweight individuals [38]. Moreover, higher adherence to the Meat DP was also associated with higher energy intake, another influencing factor for TG levels [37]. However, although the percentage of overweight participants was high, no difference was observed between nutritional status, as indicated by BMI and waist/height ratio, and adherence to the Meat DP.

The ratio TG/HDLc is used as a predictor of CVD, as an increased value indicates a LDLc phenotype of small dense particles with strong atherogenic properties [24]. A higher TG/HDLc ratio has been previously associated with male sex [39]. Consistently, in our study, participants with highest adherence to the Meat DP were more likely to be men. In addition, tobacco consumption, another lifestyle risk factor for CVD was also higher in those at the highest tertile compared with the lowest tertile. Physical activity, a protective behavioral factor for the reduction of cardiovascular disease [40] remained similar across tertiles of the Meat DP. In our study, a large proportion of the participants had low levels of physical activity, overreaching previous estimations that reported a prevalence of low physical activity of 22.8% [7] for both sexes.

When examined in subgroups by age categories, the significant associations between lipoproteins phenotype and adherence to Meat DP observed in the whole group were evident only in younger participants, additionally to higher proteinuria, which would indicate the presence of incipient vascular risk factors. The lack of association in older participants may be related to the use of lipid-lowering medication and/or to the loss of statistical power due to the small sample size of the age subgroups.

The Prudent DP identified in our study was associated with 78% reduced risk of vitamin D deficiency. There is evidence that dietary vitamin D intake has a minor influence in serum levels in cases where sunlight exposure is sufficient [41]. Montevideo, at latitude 35° S experiences in summer (December–February) an average of 13–15 h of daylight and during winter (June–August), an average of 9–11 h of daylight. It is worth noting that in Uruguay, the routine fortification of food with vitamin D is not frequent, thus a majority of vitamin D input probably comes from sun exposure. A higher calcium intake was observed in participants with higher adherence to the Prudent DP, reaching an almost adequate mineral intake [42]. Calcium intake appears to be important in regulation of vitamin D levels [43], which could be consistent with our results. Moreover, our finding could be related to better sunlight exposure from more time spent outdoors in those with increased adherence to the Prudent PD, although we did not collect this information. On the other hand, an association with other healthier lifestyle factors, such as physical activity, was not found with higher adherence to the Prudent DP. When examined by age categories, lower odds of vitamin D deficiency with increased adherence to the Prudent DP was significant only in older participants, suggesting a benefit of the healthier DP with increased age. Although there is a lack of information in the region related to the DP and vitamin D status, previous studies in South America have reported high prevalence of vitamin D deficiency in older participants [44,45]. A previous report suggested that the modest protective effect of healthier DPs is often related to complementary health behaviors [11]. In our study, adherence to the Prudent DP was more common among participants with higher education (particularly in the older ones), similar to a previous study [34], which could reflect that they are able to better understand the importance of healthy eating. The Cereal and Mate DP identified in this study emerged with the mate infusion beverage as an important component. Mate infusion is widely consumed in Uruguay, usually in amounts that are over a liter per day. This infusion has proven many positive health properties, associated with the antioxidant and cardiovascular protective activity related to caffeine, polyphenolic compounds, and saponins [46]. In our study, no difference was observed in lipids or glycemic profile of participants following this DP characterized by mate infusion intake in contrast to a previous study in heavy mate infusion drinkers [47]. However, younger participants with highest adherence to the Cereal and Mate DP exhibited higher HDLc, compared to those with lowest adherence. This is consistent with previous evidence showing that consumption of mate infusion was associated with elevation of antioxidant enzyme paraoxonase-1, a marker of inflammation closely related to HDLc, highlighting the cardio-protective role of mate infusion [48]. On the other hand, younger participants with highest adherence to the Cereal and Mate DP had higher SBP, and were also more likely to have low physical activity and to be less educated, factors that have been reported previously to be associated with higher BP [49]. These results highlight the importance of BP screening in vulnerable populations such as those less educated, as awareness of high BP is low in our region [50]. Two characteristically eating features of heavy mate infusion drinkers were observed in our study. On the one side, the frequent substitution of milk intake by mate, which could explain the lower calcium intake. Previous reports showed that in Uruguay, the calcium intake from dairy foods predicts the total calcium intake [51]. On the other side, the higher consumption of carbohydrates-rich foods such as crackers and bread with mate [47], could be related to a higher carbohydrates and energy intake among those with the highest adherence to the Cereal and Mate DP. Consistently, participants with highest adherence to this DP had more than two times higher likelihood of being overweight compared to those with lowest adherence. In men, the magnitude rose to more than 6 times higher likelihood of being overweight, but no association was found when analyzed for excessive weight including overweight and obesity categories. A previous study that identified a “Starchy-Sugar” DP (characterized by high loads to bakery products and refined grains-starchy vegetables) similar to the Cereal and Mate DP, could not find a positive association with excessive body weight [52]. Although evidence in animals and human studies had shown the protective effect of long-term consumption of mate infusion in body weight [46], a previous study has hypothesized that higher carbohydrate intake may occur as a compensatory mechanism following induced hypoglycemia, and would be responsible for higher body weight [47]. Moreover, participants with highest adherence to the Cereal and Mate DP were more likely to have low physical activity.

### 4.2. Dietary Patterns and Cardiovascular and Kidney Phenotypes

Recently, a review concluded that DPs play a key role in the relationship with age-related diseases and phenotypes, especially by following a Healthy DP, as it was associated with lower levels of biomarkers of inflammation and oxidative stress [53].

In the present study, we expected the Prudent DP to be associated with better cardiovascular phenotype; however, no significant associations were found. In contrast, previous studies in South America that characterized DP among adults have described the relation of a Prudent DP with better serum lipid profile [31,32] and lower markers of inflammation [36]. Although the latter studies studied phenotypes included in our study (such as basic lipid profile), they also included plasma concentrations of apo B and C-reactive protein, as well as soluble vascular cell adhesion molecule-1 and soluble E selectin [36], molecules that have been implicated in early stages of disease process. In our study, we assessed pulse wave velocity to quantify vascular aging but no association was found. Although in our study risk factors such as obesity, hypertension or diabetes were similar in participants following a Prudent DP, the prevalence was higher with respect to the other DP and National estimates [7]. These may reflect the improvement of diet through adoption of a healthier DP among participants with poorer metabolic health conditions. In agreement, a recent study with a representative sample of the Brazilian population [34] showed that people with accumulation of NCDs were more likely to adopt the healthy DP. In fact, in our study, fiber and protein intake (mainly from fish and vegetables) were higher, and alcohol intake lower, among those with higher adherence to the Prudent DP. Nevertheless, it should be taken into account that overall, fiber intake was below the recommendation, as vegetable and fruit consumption in Uruguay is far below the amount recommended in the current guidelines [7]. Medical counselling and prescription to adopt a healthier DP, regarding particularly the composition and amount of fat is frequent among subjects at high cardiovascular risk. However, the adoption of changes regarding fat intake are often difficult, especially due to the negative taste perceptions and family compliance [54]. That may explain why participants with higher adherence to the Prudent DP had also higher intake of all dietary fatty acids. Moreover, increasing fruits and vegetable intake, do not necessarily translate into the reduction of dietary fat intake, unless conscious efforts regarding diet are done [55].

The lack of association between the Meat DP with the cardiovascular phenotypes, considering the cumulative risk factors described previously, may be related to the protective effect of younger age, as this study included a broad age range with most participants at middle age. Although we did not find clinical manifestations of cardiovascular disease, it should be taken into consideration that these participants had higher TG and TG/HDLc ratio and these markers have recently shown to predict arterial stiffness progression [56]. Thus, they may be at an increased risk of arterial stiffness although the cumulative damage on the vascular wall are not evident yet.

In the current study, no association was detected between any of the identified DP and BP. Previous reports in Southern Brazil, only found an association between DP and hypertension in women between 40–60 years [57]. In our study, even when the analyses were performed by sex and age categories, no association were found. On the other hand, other studies found a negative association between a “Healthy” DP (such as the Prudent DP) and systolic BP, and lower occurrence of hypertension [32]. A recent longitudinal study in Brazil reported the protective effect of following a DP characterized by healthy food selection [58] in incident hypertension. Moreover, it found that body weight explained as much as 10% in the reduction risk of high BP. In our study, participants following the Prudent DP had high prevalence of obesity, which may partially explain the lack of association with the reduced risk of hypertension. Regarding kidney variables, highest adherence to the Prudent DP among younger participants was associated with lower eGFR. Although it is not possible to infer causality due to the transversal design of our study, the latter association may derive from reverse causality, often overlooked in associations between risk factors and adverse health outcomes [59]. In this sense, younger participants with lower eGFR could be following a Prudent DP since they have a greater disease burden. Accordingly, a recent study assessed the relation between DP and renal function, and found that lower eGFR was more frequent in those who followed a “fruit and vegetable” DP [60]. In our study, no association was found between DP and nephrolithiasis, although a previous study showed that environmental factors such as diet, fluid intake, and BMI can influence kidney stone formation [61]. Dietary protein has been associated with higher risk of kidney stone formation, thus we expected to find an association among those with higher adherence to the Meat DP. In contrast, we found no association, which could be related to the fact that nephrolithiasis was assessed by the self-report.

Although case-control studies on cancer have associated cancer risk with DP in Uruguay [33,35], no population-based studies characterizing DP in Uruguay have been reported. Our study characterized local DP in adults in Uruguay in relation to multiple outcomes of cardiovascular and kidney phenotypes under ordinary life contexts. Some limitations, however, should also be noted. First, as our study was performed in a geographically defined area from Montevideo, findings cannot be generalized to the whole country. Additionally, causality cannot be established due to the cross-sectional study analysis. Moreover, reverse causality might be present reflecting our contradictory results between the Prudent DP and kidney phenotypes. Second, there may have been underreports in the self-assessment of dietary intake. In our study, obesity rates were higher than National estimates [7] thus the lack of heterogeneity in nutritional status might be present. Moreover, participants with higher BMI tend to under-report energy intake [62]. Furthermore, the proportion of women in this study was high, introducing a potential bias in the interpretation of the results. It should be also taken into consideration that DP were empirically derived through PCA and the three DP combined only explained in part the variation in the diet. Another limitation was the small sample size for the sub-analyses by age categories and sex. Since the sample for this analysis was based on the availability of dietary information, a potential introduction of the response bias may apply, as the subjects included had higher BMI, pulse wave velocity, higher proportion of women, and less proportion of CKD than the subjects not-included.

## 5. Conclusions

We identified three DP: Meat, Prudent, and Cereal and Mate in an adult population cohort from a geographically defined area in Montevideo, Uruguay. The Meat DP was more common among younger participants; the Prudent DP among older participants with higher education; and the Cereal and Mate was more common among participants with unhealthy lifestyle behaviors. The traditional Meat DP was associated with worse blood lipid profile; the Prudent DP was associated with reduced risk of vitamin D deficiency, and the Cereal and Mate DP with higher odds of excessive body weight. No direct associations of dietary patterns with hypertension, arterial stiffness, chronic kidney disease, and nephrolithiasis were found in the studied population, nor by age categories or sex. The Cereal and Mate DP emerged as a local dietary pattern that warrants further investigation.

## Figures and Tables

**Figure 1 nutrients-13-02213-f001:**
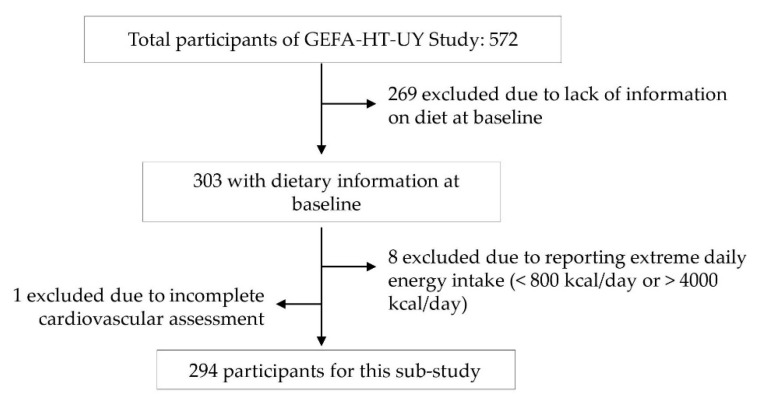
Flow chart of participants. GEFA-HT-UY: Genotype Phenotype and Environment of Hypertension in Uruguay Study.

**Figure 2 nutrients-13-02213-f002:**
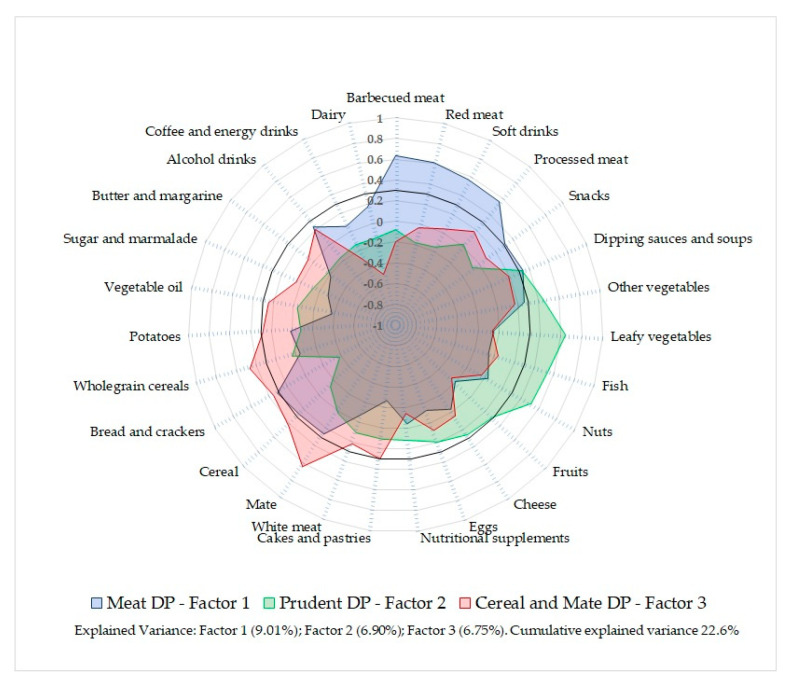
Factor-loading plot characterizing the three dietary patterns derived from the 27 food items in Uruguayan adults. DP: Dietary pattern.

**Table 1 nutrients-13-02213-t001:** Factor-loading matrix characterizing the three dietary patterns derived from the 27 food items in Uruguayan adults ^1^.

Description			
Original Food Groups	Factor 1 Meat DP	Factor 2 Prudent DP	Factor 3 Cereal and Mate DP
Barbecued meat	0.63		
Red meat	0.61		
Soft drinks	0.57		
Processed meat	0.55		
Snacks	0.31		
Dipping sauces/soups	0.34	0.33	
Other vegetables		0.44	
Leafy vegetables		0.64	
Fish		0.54	
Nuts		0.51	
Fruits			
Cheese			
Eggs			
Nutritional supplements			
Cakes and pastries			0.30
White meat			
Mate			0.63
Cereal			0.42
Bread and crackers	0.31	−0.38	0.36
Wholegrain cereals			0.47
Potatoes			
Vegetable oil	−0.38		
Sugar/marmalade			
Butter/margarine			
Alcohol drinks			
Coffee/energy drinks			
Dairy			−0.50
Explained variance (%)	9.01	6.90	6.75

^1^ Factor loadings represent the correlation between factor scores and intakes of food items. Positive factor loadings < 0.30 and negative factor loadings > −0.30 were omitted in the table for simplicity. DP: Dietary pattern.

**Table 2 nutrients-13-02213-t002:** Characteristics of dietary intake according to tertiles (T) of dietary patterns in Uruguayan adults.

	Meat DP				Prudent DP				Cereal and Mate DP			
	T1	T2	T3	*p* *	T1	T2	T3	*p* *	T1	T2	T3	*p* *
Total energy intake (kcal/day)	1633	1596	2124	<0.0001	1893	1710	1751	0.13	1599	1624	2125	<0.0001
Protein (% energy)	15.2	16.4	16.7	0.0058	14.8	15.9	17.6	<0.0001	16.9	15.8	15.6	0.019
Fat (% energy)	42.2	41.3	40.3	0.065	38.7	41.4	43.6	<0.0001	42.3	42.2	39.3	0.0043
SFA (% energy)	11.7	11.0	11.6	0.83	10.9	11.6	12.0	0.023	11.4	11.4	11.6	0.80
MUFA (% energy)	16.7	15.2	15.1	0.026	14.3	15.7	17.0	<0.0001	16.1	15.9	15.0	0.11
PUFA (% energy)	11.0	11.5	9.8	0.039	10.1	11.0	11.1	0.076	11.1	11.5	9.8	0.019
Cholesterol (mg/1000 kcal)	101.1	95.9	101.4	0.97	76.7	93.3	127.9	<0.0001	99.4	95.2	103.7	0.61
Carbohydrate (% energy)	42.6	42.3	43.0	0.74	46.5	42.7	38.8	<0.0001	40.7	42.0	45.2	0.0011
Fiber (g/1000 kcal)	8.8	8.3	7.1	0.0006	7.2	8.0	9.0	0.0004	8.0	7.3	8.9	0.085
Sodium (mg/day)	1116.5	1408.2	2340.8	<0.0001	1715.2	1604.9	1549.9	0.28	1358.0	1389.9	2113.2	<0.0001
Calcium (mg/day)	768.6	810.2	825.7	0.39	718	768	917	0.0025	951	750	705	0.0002
Alcohol (g/day)	8.0	12.8	27.5	<0.0001	26.8	13.9	13.4	0.0030	15.8	20.1	18.2	0.63

Total *n* = 294. * *p*-Value for trend. DP: Dietary pattern; SFA: Saturated fatty acids; MUFA: Monounsaturated fatty acids; PUFA: Polyunsaturated fatty acids; T3: Participants with highest adherence to dietary pattern; T1: Participants with lowest adherence to dietary pattern.

**Table 3 nutrients-13-02213-t003:** Demographic and anthropometric characteristics, conventional risk factors, and cardiovascular and renal phenotypes by tertiles (T) of dietary patterns in Uruguayan adults.

	Meat DP				Prudent DP				Cereal and Mate DP			
	T1	T2	T3	*p* *	T1	T2	T3	*p* *	T1	T2	T3	*p* *
*n*	97	99	98		97	98	99		98	97	99	
Female, *n* (%)	74 (76.3)	68 (68.7)	57 (58.2)	0.0068	62 (63.9)	66 (67.4)	71 (71.7)	0.24	63 (64.3)	77 (79.4)	59 (59.6)	0.48
Age, (years) ^ǂ^	60.4 ± 14.4	53.8 ± 16.5	44.5 ± 15.5	<0.0001	49.5 ± 18.6	54.0 ± 15.9	55.1 ± 15.2	0.018	52.0 ± 17.9	53.8 ± 16.4	52.9 ± 16.0	0.72
Education ≤9 years, *n* (%)	66 (68.8)	54 (54.6)	68 (69.4)	0.91	68 (70.1)	66 (67.4)	54 (55.1)	0.029	52 (53.6)	66 (68.0)	70 (70.7)	0.013
Low physical activity, *n* (%)	75 (77.3)	73 (73.7)	74 (75.5)	0.77	75 (77.3)	75 (76.5)	72 (72.7)	0.45	64 (65.3)	79 (81.4)	79 (79.8)	0.018
Current smokers, *n* (%)	7 (7.5)	18 (19.6)	30 (33.3)	<0.0001	17 (17.7)	17 (19.5)	21 (22.8)	0.38	11 (12.0)	19 (20.7)	25 (27.5)	0.009
Alcohol use, *n* (%)	28 (30.1)	26 (28.3)	43 (47.8)	0.013	31 (32.3)	32 (36.8)	34 (37.0)	0.50	29 (31.2)	25 (27.5)	43 (47.3)	0.023
Obesity, *n* (%)	39 (40.2)	40 (40.4)	37 (37.8)	0.73	37 (38.1)	48 (49.0)	31 (31.3)	0.32	39 (39.8)	40 (41.2)	37 (37.4)	0.73
Hypertension, *n* (%)	46 (47.4)	40 (40.4)	21 (21.4)	0.0002	28 (28.9)	41 (41.8)	38 (38.4)	0.17	32 (32.7)	40 (41.2)	35 (35.4)	0.70
Uncontrolled HT-OBP, *n* (%)	27 (27.8)	23 (23.2)	10 (10.2)	0.0022	15 (15.5)	22 (22.5)	23 (23.2)	0.18	18 (18.4)	22 (22.7)	20 (20.2)	0.75
Uncontrolled HT-ABPM, *n* (%)	21 (29.6)	16 (25.4)	20 (33.9)	0.62	16 (28.1)	19 (28.4)	22 (31.9)	0.63	15 (26.3)	20 (30.3)	22 (31.4)	0.54
Arterial stiffness, *n* (%)	22 (31.4)	17 (28.3)	5 (11.4)	0.022	7 (16.3)	22 (39.3)	15 (20.0)	0.98	11 (21.2)	18 (30.5)	15 (23.8)	0.79
Pulse wave velocity (m/s) ^ǂ^	9.2 ± 2.9	9.2 ± 3.1	8.0 ± 2.7	0.061	7.9 ± 2.8	9.7 ± 3.2	8.8 ± 2.6	0.23	8.5 ± 2.7	8.9 ± 2.8	9.2 ± 3.2	0.23
Diabetes, *n* (%)	15 (15.5)	11 (11.1)	5 (5.1)	0.019	8 (8.3)	11 (11.2)	12 (12.1)	0.38	9 (9.2)	11 (11.3)	11 (11.1)	0.66
Proteinuria (g/24 h) ^ǂ^	0.14 (0.16)	0.12 (0.063)	0.17 (0.10)	0.13	0.15 (0.083)	0.15 (0.17)	0.13 (0.08)	0.26	0.13 (0.10)	0.15 (0.16)	0.14 (0.08)	0.95
ACR (mg/g) ^ǂ^	12.5 ± 43.9	6.6 ± 10.8	10.7 ± 24.9	0.90	6.1 ± 16.0	12.2 ± 37.2	12.2 ± 25.7	0.23	9.1 ± 34.6	7.7 ± 15.7	12.4 ± 26.9	0.56
CKD, *n* (%)	3 (8.6)	3 (5.9)	5 (7.3)	0.87	2 (3.2)	5 (9.4)	4 (10.0)	0.16	2 (3.5)	5 (10.4)	4 (8.0)	0.35
Nephrolithiasis, *n* (%)	13 (13.4)	12 (12.1)	12 (12.2)	0.81	14 (14.4)	12 (12.2)	11 (11.1)	0.48	13 (13.3)	14 (14.4)	10 (10.1)	0.50

Total *n* = 294. * *p*-Value for trend. ^ǂ^ Mean values ± SD. DP: Dietary pattern; HT: Hypertension; OBP: Office blood pressure; ABPM: Ambulatory blood pressure monitoring (*n* = 193); ACR: Urinary albumin to creatinine ratio; CKD: Chronic kidney disease. Low physical activity was defined as energy expenditure < 600 mets/min per week. Current smokers (yes/no). Alcohol use was considered if reported drinking alcohol at least once a week (yes/no). Obesity was defined as BMI ≥ 30 kg/m^2^. Hypertension was defined as blood pressure (BP) of at least 140 mm Hg systolic or 90 mm Hg diastolic or use of antihypertensive drugs. Uncontrolled hypertension by office measurement was defined as BP of at least 140 mm Hg systolic or 90 mm Hg diastolic. Uncontrolled hypertension by ABPM measurement was defined as BP of at least 130 mm Hg systolic or 80 mm Hg diastolic. Pulse wave velocity was the distance in meters divided by the transit time in seconds (*n* = 174). Arterial stiffness was defined as a pulse wave velocity > 10 m/s. Diabetes was defined as self-reported diagnosis, a fasting plasma glucose of 126 mg/dl or higher or use of antidiabetic drugs. CKD was defined as an eGFR < 60 mL/min/1.73 m^2^ or an albumin/creatinine ratio > 30 mg/g, based on a single determination (*n* = 155). Nephrolithiasis was defined according to the report (yes/no).

**Table 4 nutrients-13-02213-t004:** Median metabolic, anthropometric, and cardiovascular and renal variables by tertiles (T) of dietary patterns in Uruguayan adults.

	Meat DP				Prudent DP				Cereal and Mate DP			
	T1	T2	T3	*p* *	T1	T2	T3	*p* *	T1	T2	T3	*p* *
*Metabolic characteristics*												
Fasting glucose (mg/dL)	93.3	92.6	98.6		91.9	91.0	101.3		93.7	95.8	95.9	
Adjusted median difference	Ref	0.57	1.82	0.49	Ref	1.72	4.48	0.097	Ref	0.055	0.14	0.96
Total cholesterol (mg/dL)	208.2	207.2	203.0		201.1	206.2	208.9		208.0	205.3	204.2	
Adjusted median difference	Ref	−0.53	−1.70	0.69	Ref	2.23	5.81	0.17	Ref	0.070	0.17	0.97
Triglyceride (mg/dL)	139.0	131.8	155.8		134.9	154.2	138.4		153.8	133.6	139.9	
Adjusted median difference	Ref	5.23	16.68	0.040	Ref	0.42	1.08	0.90	Ref	0.40	0.99	0.91
LDLc (mg/dL)	127.6	128.0	126.6		123.8	125.2	131.9		127.5	127.7	126.9	
Adjusted median difference	Ref	−0.24	−0.78	0.82	Ref	2.42	6.31	0.075	Ref	−0.39	−0.98	0.80
HDLc (mg/dL)	49.6	52.1	46.0		51.6	47.0	49.3		48.3	49.5	49.6	
Adjusted median difference	Ref	−1.09	−3.48	0.012	Ref	−0.24	−0.62	0.66	Ref	0.78	1.95	0.20
Non-HDL colesterol (mg/dL)	158.1	154.8	157.1		149.7	158.0	160.0		158.9	155.2	155.2	
Adjusted median difference	Ref	0.36	1.13	0.78	Ref	2.60	6.79	0.10	Ref	−0.43	−1.07	0.81
TG/HDLc ratio	3.19	2.97	3.80		3.01	3.72	3.25		3.62	3.12	3.27	
Adjusted median difference	Ref	0.22	0.70	0.0031	Ref	0.0082	0.021	0.93	Ref	−0.051	−0.13	0.63
HOMA-r	2.83	2.87	3.57		2.61	2.96	3.69		2.95	3.38	3.13	
Adjusted median difference	Ref	0.17	0.54	0.044	Ref	0.17	0.44	0.10	Ref	0.0099	0.025	0.93
Neut/Lymph ratio	1.91	1.90	1.89		1.87	1.99	1.84		1.83	1.99	1.89	
Adjusted median difference	Ref	0.045	0.14	0.096	Ref	−0.009	−0.023	0.79	Ref	0.057	0.14	0.13
Serum 25(OH)D (ng/mL)	26.9	23.9	25.8		25.6	24.3	26.2		24.7	26.8	25.0	
Adjusted median difference	Ref	−0.31	−1.00	0.27	Ref	0.35	0.92	0.32	Ref	0.32	0.80	0.43
*Anthropometric characteristics*												
BMI (kg/m^2^)	27.3	28.2	28.2		27.1	29.2	27.6		28.0	27.6	28.2	
Adjusted median difference	Ref	0.22	0.69	0.29	Ref	0.18	0.46	0.48	Ref	−0.029	−0.071	0.92
Waist/height ratio	0.58	0.58	0.59		0.59	0.59	0.58		0.59	0.58	0.59	
Adjusted median difference	Ref	0.0026	0.0084	0.10	Ref	−0.002	−0.004	0.40	Ref	0.0015	0.0037	0.50
*Cardiovascular and renal characteristics*												
Systolic BP (mm Hg)	124.2	125.4	124.9		123.9	124.5	126.1		124.6	124.1	125.8	
Adjusted median difference	Ref	−0.21	−0.66	0.67	Ref	0.18	0.47	0.77	Ref	1.30	3.24	0.059
Diastolic BP (mm Hg)	81.2	82.0	80.2		80.1	80.2	82.8		81.0	81.1	81.2	
Adjusted median difference	Ref	−0.29	−0.91	0.34	Ref	0.44	1.14	0.24	Ref	0.42	1.03	0.32
ABPM Systolic BP (mm Hg)	118.7	121.9	123.2		122.7	122.0	120.3		121.3	119.7	122.4	
Adjusted median difference	Ref	0.66	2.08	0.20	Ref	−0.57	−1.53	0.37	Ref	1.65	3.73	0.021
ABPM Diastolic BP (mm Hg)	75.2	76.1	77.9		76.4	76.6	76.4		76.0	75.1	77.4	
Adjusted median difference	Ref	0.20	0.62	0.62	Ref	0.084	0.23	0.86	Ref	1.43	3.22	0.010
Pulse wave velocity (m/s)	8.55	9.32	8.93		8.05	9.49	8.98		8.79	8.75	9.17	
Adjusted median difference	Ref	0.12	0.33	0.38	Ref	0.010	0.025	0.94	Ref	0.050	0.13	0.74
Serum creatinine (mg/dl)	0.82	0.81	0.85		0.81	0.85	0.83		0.83	0.82	0.84	
Adjusted median difference	Ref	0.0013	0.0041	0.82	Ref	0.0076	0.020	0.29	Ref	0.0048	0.012	0.56
Proteinuria (g/24-h)	0.15	0.12	0.17		0.15	0.15	0.13		0.14	0.16	0.13	
Adjusted median difference	Ref	0.0054	0.017	0.26	Ref	−0.003	−0.008	0.57	Ref	−0.0004	−0.001	0.95
ACR (mg/g)	14.9	7.3	15.6		9.3	14.0	15.3		11.6	10.0	17.0	
Adjusted median difference	Ref	0.081	0.25	0.95	Ref	095	2.65	0.55	Ref	2.28	6.04	0.19
eGFR (mL/min/1.73 m^2^)	95.5	95.9	92.1		96.0	93.2	94.1		94.2	94.9	93.9	
Adjusted median difference	Ref	−0.30	−0.97	0.49	Ref	−0.77	−2.02	0.16	Ref	−0.17	−0.43	0.78

Total *n* = 294. * *p*-Value for linear trend. Ref: Reference value; LDLc: Low-density lipoproteins cholesterol; HDLc: High-density lipoproteins cholesterol; TG/HDLc: Triglycerides/high-density lipoproteins cholesterol ratio; HOMA-r: Homeostasis Model Assessment of Insulin Resistance; Neut/Lymph: Neutrophils/lymphocytes; 25(OH)D: 25-Hydroxy-vitamin-D; BMI: Body mass index; BP: Blood pressure; ABPM: Ambulatory blood pressure monitoring (*n* = 193); ACR: Urinary albumin to creatinine ratio; eGFR: Estimated glomerular filtration rate, derived from the Chronic Kidney Disease Epidemiology Collaboration equation. Pulse wave velocity (*n* = 174), and albumin/creatinine ratio (*n* = 155). Multivariable models were adjusted for continuous variables: Age, BMI, daily energy intake and for categorical variables: Sex, smoking status (current smoking: Yes/no), physical activity (inactive <600 mets/min per week, active ≥600 mets/min per week), and education (≤9, >9 years). For 25(OH)D, models were additionally adjusted for season (summer, autumn, winter, spring), and for office blood pressure for use of antihypertensive drugs (yes/no).

**Table 5 nutrients-13-02213-t005:** General adjusted odds ratio (OR), 95% confidence intervals (CI) for cardiovascular and renal risk phenotypes according to tertiles (T) of dietary patterns in Uruguayan adults.

Variables	Meat DP					Prudent DP					Cereal and Mate DP				
	T1	T2		T3		T1	T2		T3		T1	T2		T3	
	Ref	OR	95% CI	OR	95% CI	Ref	OR	95% CI	OR	95% CI	Ref	OR	95% CI	OR	95% CI
Hypertension (Office SBP ≥ 140 mm Hg or DBP ≥ 90 mm Hg or use of antihypertensive drugs)															
Adjusted	1.00	2.14	0.69–6.62	0.37	0.08–1.81	1.00	1.64	0.48–5.62	1.97	0.61–6.41	1.00	1.34	0.44–4.04	0.65	0.18–2.30
Uncontrolled hypertension by office measurement (SBP ≥ 140 mm Hg or DBP ≥ 90 mm Hg)															
Adjusted	1.00	1.19	0.57–2.47	0.48	0.18–1.31	1.00	1.35	0.59–3.06	1.67	0.75–3.74	1.00	1.08	0.49–2.38	0.96	0.42–2.19
Uncontrolled hypertension by ABPM measurement (SBP ≥ 130 mm Hg or DBP ≥ 80 mm Hg)															
Adjusted	1.00	0.95	0.41–2.18	1.83	0.69–4.82	1.00	0.81	0.33–1.97	1.00	0.43–2.35	1.00	1.26	0.53–3.00	0.95	0.40–2.28
Arterial stiffness (pulse wave velocity > 10 m/s)															
Adjusted	1.00	2.86	0.84–9.80	2.14	0.46–9.99	1.00	4.05	0.97–16.93	1.76	0.49–6.34	1.00	0.95	0.28–3.27	0.55	0.14–2.26
High TG/HDLc ratio (>3.75 for men and >3.0 for women)															
Adjusted	1.00	0.93	0.47–1.82	1.11	0.51–2.39	1.00	1.46	0.76–2.79	0.85	0.43–1.67	1.00	0.55	0.28–1.07	0.72	0.36–1.43
Overweight (BMI ≥ 25 < 30 kg/m^2^)															
Adjusted	1.00	1.30	0.69–2.43	0.96	0.46–1.99	1.00	1.16	0.62–2.18	1.42	0.77–2.65	1.00	2.14	1.11–4.12	2.76	1.40–5.47
Overweight and Obesity (BMI > 25 kg/m^2^)															
Adjusted	1.00	1.82	0.84–3.95	1.01	0.43–2.36	1.00	3.12	1.39–7.02	1.33	0.66–2.68	1.00	2.26	1.10–4.65	2.48	1.13–5.43
High waist-to-height ratio (≥0.5)															
Adjusted	1.00	0.53	0.15–1.89	1.43	0.36–5.74	1.00	0.84	0.21–3.32	0.66	0.20–2.14	1.00	0.55	0.16–1.84	1.64	0.38–6.99
Vitamin D deficiency (serum 25(OH)D < 12 ng/mL)															
Adjusted	1.00	4.82	1.32–17.65	3.07	0.60–15.66	1.00	0.64	0.20–2.02	0.22	0.048–0.99	1.00	0.74	0.23–2.38	0.72	0.20–2.60
Chronic kidney disease (eGFR < 60 mL/min/1.73 m^2^ or ACR > 30 mg/g)															
Adjusted	1.00	0.64	0.11–3.75	1.31	0.21–8.27	1.00	2.96	0.49–17.68	3.61	0.56–23.48	1.00	3.33	0.54–20.64	2.94	0.40–21.61
Nephrolithiasis (yes)															
Adjusted	1.00	0.98	0.38–2.54	1.74	0.58–5.28	1.00	0.61	0.24–1.51	0.59	0.23–1.53	1.00	1.52	0.61–3.80	0.88	0.32–2.39

Total *n* = 294. Ref: Reference values; SBP: Systolic blood pressure; DBP: Diastolic blood pressure; ABPM: Ambulatory blood pressure monitoring (*n* = 193); TG: Triglycerides; HDLc: High-density lipoproteins cholesterol; BMI: Body mass index; 25(OH)D: 25-Hydroxy-vitamin-D; eGFR: Estimated glomerular filtration rate; ACR: Urinary albumin to creatinine ratio. Arterial stiffness was defined as a pulse wave velocity > 10 m/s (*n* = 174). Chronic kidney disease was defined as an eGFR < 60 mL/min/1.73 m^2^ or an ACR > 30 mg/g based on a single determination (*n* = 155). Nephrolithiasis was defined according to the report (yes/no). Models adjusted for continuous variables: Age, BMI, daily energy intake and categorical variables: Sex, smoking status (current smoking: Yes/no), physical activity (inactive < 600 mets/min per week, active ≥ 600 mets/min per week), and education (≤9 years, >9 years). For 25(OH)D, they were additionally adjusted for season (summer, autumn, winter, spring), and for BP for use of antihypertensive drugs (yes/no).

## Data Availability

Datasets are not available for the public domain according to the national legislation and may be available upon request to the corresponding author.

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
