# Peer review of "Association of Dietary Patterns with Cardiovascular and Kidney Phenotypes in an Uruguayan Population Cohort"

_nutrients, 2021, doi:10.3390/nu13072213_

Round 1

Reviewer 1 Report

Thank you for giving me the opportunity to read this interesting manuscript from Uruguay. Moliterno et al present a well-conducted study among ca 300 randomly found individuals in the city of Montevideo. The aim of the study was to identify major dietary (DP) patterns and to assess associations with anthropometric characteristics, metabolic, cardiovascular and kidney phenotypes. Three DP identified. They found that meat DP was associated with higher TG, HDLc, TG/HDLc ratio and HOMAr. Prudent DP was associated with reduced risk of vitamin D deficiency. Cereal and Mate DP was associated with increased risk of excessive body weight. The introduction is ok, maybe a bit too long. The methods are presented very well. The results presentation can be improved. The discussion has some potential for improvement as well, as I found it too diffuse I n some parts.

Moliterno et al have done extensive analyses, and this study could be presented in 2 or 3 different manuscripts.  The authors, however, choose to present everything in one. Nonetheless, I believe that this manuscript will be good for publication after some minor changes/improvements. I present my comments for consideration bellow.

Abstract:

  • I think the phrase “We aimed to identify major dietary patterns (DP) in a population adult cohort in Uruguay…” is grammatically not correct and that affect the meaning and the flow of the reading. Please consider the expressions “...in an adult cohort in Uruguay” or “...in an adult population in Uruguay” or “…in an adult population sample in Uruguay”.
  • I don’t understand the sentence: “Three DP were identified (Meat, Prudent, Cereal and Mate) explaining 22.6% of total variance.”. What variance do the authors mean?
  • In the last sentence of the abstract, I believe that the authors mean “in the studied population” or “in the whole studied population” and not “in the general population”.

Introduction:

  • In the introduction the authors state “In most countries of Latin America, cardiovascular risk factors are highly prevalent [2], and particularly the phenotypes related to unhealthy lifestyle and dietary transition [3].”. I find this a little confusing. First cardiovascular factors are highly prevalent, that is ok. But thereafter, the authors talk about phenotypes. Is risk factors and phenotypes the same think? In that case it would be wise to stick in one term through the whole text. That line means that there are phenotypes (or cardiovascular risk factors) related to unhealthy diet. I believe, after reading the title, that this is what this study wants to examinate. The authors need to explain more about what is known in the introduction. However, I am not very familiar with the term “cardiovascular phenotype”. I neither know what dietary transition is, the authors choose that term instead of dietary pattern, why? It can be better if the authors skip dietary transition in the introduction, I did not find it useful when I read the whole text either.
  • “The theory of the nutrition transition posits that dietary changes relate to the complex interplay of fluctuations in patterns of agricultural, health, and socioeconomic factors, among others [5].”, I found those lines very difficult to understand. Please consider reformulation since many of the readers would spend a lot of time reading again those lines. Do the authors need to discuss dietary transition in the introduction? I suggest to remove this part. The introduction is already a bit too long.
  • “Overweight and obesity in adult population, a major metabolic risk factor, has increased from 57% to 65% from 2006 to 2013”. I propose to state a major risk factor of what (cardiovascular diseases?) or to skip the “a major metabolic risk factor”.
  • “Dietary patterns (DP) have been widely explored to study the diversity in diet, as they allow an holistic approach to the multidimensional dietary phenomenon [12].”, it is for me difficult to understand what the authors mean here.
  • As in the abstract; please consider to write “...in an adult cohort in Uruguay” or “...in an adult population in Uruguay” or “…in an adult population sample in Uruguay” instead of “in a population adult cohort in Uruguay”.

Population:

  • Please state when (time period) this study is conducted.
  • How were those randomly recruited nuclear families been found? Through the primary health centres? Through door knocking in the neighbourhood? Through a population register? Were they contacted through a letter send home or did they answer to an advertisement?
  • Do we know anything abut the socioeconomic status of the studied population?
  • “Dietary information using a food frequency questionnaire (FFQ) adapted was assessed by interview by a trained professional.”, maybe better to rephrase this line. I suggest, for instance, “Dietary information collected through interview by a trained professional using a frequency questionnaire (FFQ).”

Dietary Pattern Analysis:

Needs to be assessed by a person with better statistical knowledge than me.

Blood Pressure Measurements

Have all the participants been examined with ABPM or only those with hypertension diagnosis?

Statistical methods

On page 6, “according to cardiovascular risk phenotypes as follows: hypertension and uncontrolled hypertension”, I that case the term “untreated hypertension” is preferrable than “hypertension”.

Results

The characteristics of the participants are very important and need to be presented in as table 1 and not as supplementary material S1b. (Table S1a. Characteristics of participants according to age categories can stay as supplementary material). Besides, the authors do not have to repeat everything in the main text as this information is presented in table-form.

Table 2

Please state beneath the table that the lowest tertile represents the participants with poor adherence to the DP and the highest tertile the participants with best adherence. T1 lowest and T3 highest?

Table 5

No need to present the p-value, it is more than enough with the confidence interval. Please check if the p-value of 0.0059 for Overweight and Obesity in prudent T2 is correct, it does not seem to be. But, as I said, the authors should skip p-values in this table.

Discussion

  • Very good choice to start the discussion with the main findings. However, I do not believe that the sub-analyses of younger individuals belong here. Only 20 individuals among the younger group had high blood pressure, that requires some cautious when the results are discussed. The eGFR in the younger prudent group was sign lower but still very normal (106.8 mL/min/1.73 m2). Thus, we have a statistical sign difference without any clinical significance. I suggest those results not to be discussed in the first paragraph among the main findings.
  • Line 100-101: “In our study, the Meat DP was associated with a blood lipid profile indicative of insulin resistance, atherosclerosis and cardiovascular disease [23].” This statement should be changes as the study didn’t show any relation with atherosclerosis and cardiovascular disease. The authors should rephrase those lines.
  • Line 131-132: “The lack of association in older participants may be related to the use of lipid-lowering medication.” The loss of statistical power due to smaller groups plays roll in the lack of association, and has to be mentioned. The same applies for the sub-analyses of the sexes.
  • Line 195-196: “In the present study, we expected the Prudent DP to be associated with better cardiovascular phenotype; however, no significant associations were found.” The authors should discuss the fact that the studied population is relatively young (mean age ca 52) and therefore they couldn’t find manifestations of cardiovascular diseases, but precursors such as insulin resistance. However, that fact can not explain everything (e.g., no difference in hypertension). I am not sure if it is that the authors try to say in lines 221-223, it has to pointed-out clearer. Maybe the authors should address this population as a young middle-aged population in Uruguay instead of just population in Uruguay.

Reviewer 2 Report

I have reviewed the manuscript “Association of Dietary Patterns with Cardiovascular and Kidney Phenotypes in a Uruguayan Population Cohort”. Due to the scarcity of similar studies in the particular region, I think that the data and study aim of this manuscript is relevant. The manuscript is very-well written, organized and easy to follow. The study does show some odd results, but indeed some contradictory results may be encountered due to reverse causality, hence I think the interpretation of the results provided in the discussion section is overall correct for current version of the manuscript and analyses. The most important limitations regarding the study design have been acknowledged. I do have, however, some methodological concerns that warrant revision, as follows:

-is the defined area in which the study was performed representative of Uruguayan population?

-please report the participation rate (subjects invited to participate/enrolled)

-it is not clear to me how many of the participants of the study were considered participants for the present analyses. Please provide a flowchart starting with the number of subjects invited to participate in the Genotype Phenotype and Environment of Hypertension Study-GEFA-HT-UY study.

-depending on the participation rate and percentage of subjects included in the present analyses (i.e. those who had nutritional information at baseline), please consider potential introduction of response (dietary habits questionnaire/FFQ) bias. Accordingly, if appropriate, please add mentioning to the discussion and limitations paragraph.

-was it necessary to categorize variables such as TG/HDLc ratio, HDLc, waist-to-height ratio, vitamin D, etc? Although the authors appropriately provide references to support the use of such categorizations, it is known that information is lost when using a categorization approach like this for studying associations instead of raw continuous variables. I would suggest to provide sensitivity analyses using non-categorized variables to study its association with DP. The results could be additionally provided in supplemental material. Please report in the manuscript if significant differences are observed in this sensitivity analyses.

-please provide sample size calculation

-please also add power analyses

-please state whether a single or two-tailed p value was considered to indicate statistical significance.

-were there missing data? If not, please state so. Else, please report how missing data were handled.

-how would you explain that 2/3 of the study population are women? How could that observation influence the findings of the study?

-considering that 75% of the study population was overweight or obese, which is considerably higher than the reported rate at national level (Uruguay, Introduction section), would you consider that the study sample is a representative population?

-page 17, lines 100-101: I think this statement needs to be downturned.

Round 2

Reviewer 2 Report

Thank you for your revision of the manuscript. Some important pending issues as follow:

It seems to me that comment #1 was only addressed to me without the corresponding adaptations in the manuscript. Please take this comment into account to revise your reply and revision as follows: please state in the manuscript that the this cohort population study does not intend to be representative of the study population, early in methods section, in the limitations paragraph, and finally ponder this issue in the conclusions paragraph.

In reply to comment #4, the comparison table between subjects included and excluded, should include all available demographic characteristics, namely: education, physical activity, smoking status, alcohol use, arterial stiffness, pulse wave velocity, diabetes, proteinuria, ACR, CKD, nephrolitiasis. Only after having this complete comparison we can understand whether the relevant characteristics of this study haven influenced by response bias. 

In reply to comment #5, please note that categorical variables used in table 5; namely, arterial stiffness (>10 m/s), TG/HDLc ratio (<3.5, >3.0), overweight (instead of raw BMI), waist to height ratio (>0.5), vitamin D deficiency (instead of raw vitamin D).  BY sensitivity analyses I actually meant performing the same analyses as in table 5, but now using raw instead of categorical variables, whenever possible. 

In reply to comment #8, please add supporting literature for this way of proceeding.
